# The role of health literacy in the relationship between mothers' knowledge and practices of iron supplementation in children (aged 12 to 24 months): A structural equation model

**Ghazal Afshari[1], Shabnam Omidvar[ID][2]\*, Mohammadreza Kordbagheri[3]**

**1** Student of Pharmacy, School of Pharmacy, Ayatollah Amoli Branch, Islamic Azad University, Amol, Iran, **2** Social Determinants of Health Research Center, Health Research Institute, Babol University of Medical Sciences, Babol, Iran, **3** Department of Statistics, Faculty of Mathematical Sciences, Shahid Beheshti University, Tehran, Iran

\* shomidvar@yahoo.com

## Abstract

### Introduction

Iron deficiency anemia represents the most common form of anemia globally and constitutes a significant public health concern, particularly in developing nations. Therefore, supplementation is one of the best strategies for protecting children from anemia. The objective of the study was to assess the level of knowledge and practices of mothers with children aged 12 to 24 months and to assess the mediating role of health literacy in this relationship.

### Methods and materials

This cross-sectional study was conducted on 435 mothers of children (aged 12 to 24 months) referred to Tehran healthcare centers. Information was collected through socio-demographic and reproductive checklists, knowledge and practice questionnaires, and health literacy questionnaires. The data were analyzed by SPSS26 and AMOS24 software and a significance level less than 0.05 was considered.

### Results

Among the participants, 18.4% had poor knowledge, 47.4% had moderate knowledge, and only 34.2% had good knowledge. The mothers' practice score regarding iron drop feeding was moderate (8.22±2.27). A total of 37.9%, 54.7%, and 7.4% had good, moderate, and poor performance, respectively. Pearson's correlation coefficient indicated a significant positive association between mothers' understanding of iron drop feeding and their corresponding practices, as well as the practices of mothers with children aged 12 to 24 months (P < 0.001, r = 0.421). According to the results, health literacy and mothers' knowledge can predict 40% of the changes in mothers' practices, which is partial or moderate.

**Data availability statement:** The datasets used and/or analyzed during the current study are not publicly available due to restrictions imposed by the Research Ethical Committee of Islamic Azad University, Amol Branch, Iran. However, the data can be made available upon reasonable request from the corresponding author or Dr. Reza Ghadimi (rezaghadimi@yahoo.com).

**Funding:** The author(s) received no specific funding for this work.

**Competing interests:** The authors have declared that no competing interests exist.

**Abbreviations:** ID: Iron deficiency; HELIA: Health literacy of Iranian adults; IDA: Iron deficiency anemia; AMOS: Analysis of Moment Structures; CMIN/DF: Chi-square/degree-of-freedom ratio; RMSEA: Root mean square error of approximation; PCFI: Parsimonious comparative fit index; GFI: Goodness-of-Fit Index; PNFI: Parsimonious Normed Fit Index; IFI: Incremental Fit Index; CFI: Comparative Fit Index.

Health literacy plays a mediating role in the relationship between mothers' knowledge and practices.

## Conclusion

Considering the effect of anemia on children's health, paying attention to mothers' health literacy as an important factor to improve their performance is essential.

## Introduction

Approximately 2 billion of the 7.2 billion people on Earth are anemic, and iron deficiency is the main cause of anemia. It is estimated to be approximately 25 to 50 percent [1,2]. Iron deficiency (ID) is a leading factor in the onset of anemia due to multiple underlying causes. Iron deficiency anemia (IDA) is recognized as the most frequently occurring type of anemia and is one of the most common diseases affecting humans, with nearly 20% of the world's population being affected [3,4]. Anemia affects 600 million children worldwide [5].

Anemia is an indicator of malnutrition as well as a health indicator. One out of every four people suffers from anemia, and pregnant women and children are the most vulnerable groups [6]. IDA affects 40%–50% of children in developing countries and 20% of children in developed countries [7,8].

Iron is a micronutrient, and its deficiency is the most common cause of anemia worldwide; moreover, iron deficiency in early life is associated with motor, cognitive, and behavioral defects [9,10]. Iron plays a role in the function of neurotransmitters in the central nervous system and is responsible for myelination [1,11,12].

Anemia is observed to be 3 to 4 times more common in developing nations compared to developed nations [13,14]. The highest incidence of anemia occurs in children aged 9-24 months due to rapid body growth and a low-iron diet [15]. According to the updated guidelines from the World Health Organization, one effective approach to address anemia, particularly when prevalence reaches 40% or more in children, is the implementation of daily iron supplementation. [16].

Health literacy refers to the capacity of individuals to acquire, comprehend, and apply health-related information to enhance their well-being [17]. The ability to navigate health information is influenced by a multitude of social and cultural factors, educational experiences, and individual expectations, all of which are crucial in shaping health outcomes. Research demonstrates that parental health literacy has a profound impact on health-related behaviors, and insufficient health literacy in parents can adversely affect the health of their offspring [18,19].

Research indicates that inadequate health literacy correlates with detrimental lifestyle choices, increased reliance on emergency medical services, hesitance to engage in preventive healthcare measures, ineffective management of chronic illnesses, and elevated rates of hospitalization and mortality [20–22].

It has been accepted that nutrition and health are important in the first two years of life. This phase represents a crucial and optimal opportunity for cognitive development, as the brain's intellectual capacity peaks prior to the age of three [23–25].

There are several factors that prevent anemia such as knowledge, and attitude regarding supplementation [26]. Moreover mother's educational status, child's age, socioeconomic status, birth rank, breastfeeding status, and residential area, were other associated factors with iron supplementation in children [27,28].

Literacy is a crucial factor influencing health, as it allows individuals to participate meaningfully in their healthcare processes and affects health outcomes. Moreover, health literacy contributes to improved overall health and well-being, reduces health disparities, and promotes resilience within both individuals and communities [29]. It also empowers people to make informed decisions regarding their health, which results in heightened engagement and effectiveness in health-related activities [30]. Health literacy, which includes both comprehension and action, is an essential element of successful preventive medicine and serves as a means to achieve greater justice and equity in society over time [30,31].

To date, many studies have been conducted to investigate factors influencing the use of iron drops in children, but still, a lack of supplementation or unsuitable use in different communities has been shown.

Therefore, this study hypothesized that health literacy plays a bridging role between knowledge about supplementation and iron supplementation practices in children aged 12 to 24 months

## Methods and materials

The cross-sectional study was conducted among mothers referring to health centers. The recruitment period started on 12.12.2022 and ended on 27.2. 2023 after the approval of the protocol by the research board of Azad University (Ayatollah Amol Branch). First, the written informed consent letter was obtained, and then the information was collected by completing the questionnaires. The sample size calculated to be 395 by Fisher's formula

$$(1999) \quad n = \frac{((Z_1 - \frac{\alpha}{2}) + (Z_1 - \beta))^2 \, pq}{e^2}$$ The parameters were chosen as follows: p = 0.56p

= 0.56p = 0.56, based on Adham's study [28]. Regarding mothers' performance in using iron supplements, q = 1 − p, $Z_{1-\alpha/2}$ = 1.96 for a 95% confidence level, $Z_{1-\beta}$ = 0.85 for 80% statistical power, and e = 7% as the margin of error to balance precision and feasibility. These choices ensure a high likelihood of detecting true effects while minimizing Type II errors. The calculated sample size of 395 provides sufficient data for subgroup analyses and robust statistical modeling. Furthermore, the study design minimized potential attrition, ensuring the reliability and validity of the results across all plans.

### Ethics approval and consent to participate

The study design was approved by the Ethics Committee of the Ayatollah Amoli Branch of Azad University, Amol, Iran (IR.IAU.AMOL.REC.1401.110). Written informed consent was obtained from all the participants. All procedures were conducted in compliance with applicable guidelines and regulations.

### Participants and settings

The present study was conducted on a sample of 395 mothers with children aged 12 to 24 months referred to the selected health centers. The subjects were collected by convenience sampling from health centers in Tehran. The inclusion criteria consisted of providing consent, being literate, having Iranian nationality, not being addicted to drugs, and having no physical or mental illness affecting the mother's usual activities. The exclusion criterion was the unwillingness to participate in the study after completing the questionnaires.

### Data collection tools and techniques

In the present study, information was collected using questionnaires including:

1. Socio-demographic and reproductive characteristics checklists: Questions including age, place of residence, income level, education level, occupation, spouse's education, age, occupation, number of pregnancies, number of abortions, mode of delivery, and number of children.

2. The questionnaire on mothers' knowledge of and practices related to feeding iron drops to their children was administered. This questionnaire contains questions about the reasons for the need to take supplements, how to take supplements, the amount needed, and their source of information for supplementation. This questionnaire was developed by the researcher for confirmation of its validity and reliability. For face and content validity, the comments and suggestions of 10 experts in medical and health area were taken and the questionnaires were corrected. Then for reliability a pilot study was conducted on 30 people with an interval of one week, and Cronbach's alpha was determined. Cronbach's alpha was 0.788 for the awareness questionnaire and 0.704 for the performance questionnaire. Questionnaires exhibit a significant degree of reliability and internal consistency. (Supplementary file 1)

3. Health Literacy of Iranian Adults (HELIA): Montazeri et al. designed and assessed the reliability and validity of the tool. The questionnaire has 5 dimensions and 33 questions. The domains of reading, accessibility, comprehension, evaluation, decision-making and behavior were used to assess the health literacy of the Iranian population aged 18 to 65 [32].

## Statistical analysis

The data were analyzed by SPSS26 and AMOS24 software, and the significance level of the tests was considered to be less than 0.05. After confirming normality, independent parametric t-tests, analyses of variance, Pearson's correlation coefficients, and structural equation models were used for the analysis of the quantitative data. To examine the proposed research model, the SEM approach with Maximum Likelihood (ML) estimation was employed, chosen for its high efficiency in analyzing complex and indirect relationships. To assess the mediation effect, the bootstrap technique with 5,000 resamples and a 95% confidence interval was utilized. The use of bootstrap, as a non-parametric method, ensures that the distribution of the indirect effect does not rely on normality assumptions, providing more precise and reliable results.

## Results

In the present study, 435 mothers with children aged 12 to 24 months referred to the selected health centers were assessed. The average age of the participants was 29.62 ± 5.89 years, and the ages ranged from 18 to 45 years. The majority of the women participating in the study had a university education level of 56.6%, 249 individuals (67.6%) were housewives, 43.7% were employed, and 46.7% evaluated family income adequacy as moderate. A total of 43.7% of the spouses were employees. A total of 50.8 percent of their children were female. Most of the children were breastfed (59.3%).

The mothers' knowledge regarding iron drop feeding was moderate, with a mean and standard deviation of 9.53 ± 2.97. Among the participants, 18.4% had poor knowledge, 47.4% had moderate knowledge, and only 34.2% had good knowledge.

The mothers' practice score regarding iron drop feeding was moderate (8.22 ± 2.27). 37.9%, 54.7%, and 7.4% had good, moderate, and poor performance, respectively.

The analysis using Pearson's correlation coefficient indicated a significant positive relationship between the knowledge and practices of mothers regarding iron drop feeding and those of mothers with children aged 12 to 24 months (P < 0.001, r = 0.421). There was a statistically

significant relationship between mothers' employment (P = 0.008), husbands' occupation (P = 0.018) and family income adequacy (P = 0.048) and mothers' performance. No significant correlation was found between knowledge and sociodemographic characteristics (Table 1).

The results showed that the mothers had a sufficient mean health literacy score (75.02 ± 16.70). Overall, 9.9% of the mothers had insufficient health literacy, 20.4% had somehow sufficient health literacy, 34.6% had adequate health literacy, and 35.1% had excellent health literacy.

The components of health literacy exhibited the highest mean scores in the following areas: reading (79.52 ± 20.46), understanding (78.31 ± 19.51), access (75.44 ± 19.74), evaluation and decision-making (73.39 ± 20.77), and behavior (71.93 ± 18.20). Furthermore, the Pearson correlation coefficient indicated a significant relationship between mothers' knowledge and their practices related to iron supplementation and health literacy, as illustrated in Table 2.

To test the proposed model of the mediation model of health literacy in the relationship between mothers' knowledge and performance regarding iron feeding for 12- to

**Table 1. The relationships between mothers' sociodemographic characteristics, knowledge, and practices of iron supplementation in children (aged 12 to 24 months).**

| Demographic variables | | knowledge | | Practice | F/t value |
|---|---|---|---|---|---|
| | | Mean±SD | F/t value | | |
| Education | Literate | 9.29±0.53 | F=0.499 P=0.608 | 8.70±0.35 | F=0. 744 P=0. 476 |
| | intermediate school/Diploma | 9.39±0.25 | | 8.19±0.18 | |
| | Bachelor degree and higher | 9.65±0.18 | | 8.19±0.14 | |
| Occupation | Non-employed | 9.39±0.17 | F=2.320 P=0.1 | 8.04±0.13 | P=0.008 F=4.899 |
| | Freelancer job | 9.24±0.48 | | 8.08±0.35 | |
| | Employed | 10.10±0.29 | | 8.86±0.20 | |
| Spouse's education | Literate | 9.52±0.22. | t=-0.053 P=0.958 | 8.18±0.16 | t=-0.363P=0.716 |
| | intermediate school/Diploma | 9.54±0.18 | | 8.26±0.14 | |
| Spouse's occupation | Staff | 9.82±0.21 | F=1.84, P=0.159 | 8.55±0.17 | F=4.08 P=0.018 |
| | worker | 9.00±0.40 | | 7.60±0.30 | |
| | Freelancer job | 9.52±0.22 | | 8.20±0.16 | |
| Income sufficiency | Yes | 9.61±0.24 | F=2.33 P=0.098 | 8.53±0.18 | F=3.04 P=0.048 |
| | somewhat | 9.72±0.20 | | 7.59±0.15 | |
| | No | 8.88±0.32 | | 8.34±0.25 | |
| Child gender | Male | 9.51±0.21 | t=0.059 P=0.953 | 8.32±0.15 | t=1.107 P=0.269 |
| | Female | 9.50±0.18 | | 8.08±0.15 | |
| Age | | | r=-0.073, P= 0.130 | | r=0.046 P=0.339 |
| Spouse age | | | r=0.049 P=0.313 | | r=0.028, P=0.515 |

**Table 2. The standard coefficients of the health literacy model for the relationship between mothers' knowledge of and practices of iron supplementation in children (aged 12 to 24 months).**

| variables | Knowledge | Practice |
|---|---|---|
| Health literacy | r = 0.154, P = 0.001 | r = 0.140, P = 0.003 |
| Reading | r = 0.096, P = 0.046 | r = 0.066, P = 0.169 |
| Access to information | r = 0.099, P = 0.039 | r = 0.100, P = 0.036 |
| Understanding | r = 0.131, P = 0.006 | r = 0.105, P = 0.028 |
| Evaluation and decision | r = 0.113, P = 0.018 | r = 0.103, P = 0.032 |
| Behavior | r = 0.173, P < 0.001 | r = 0.169, P < 0.001 |

24-month-old children referred to selected health centers in Tehran, structural equation modeling was used.

The structural equation modeling method was used to test the proposed model of the mediating effect of health literacy on the relationship between mothers' knowledge and practices related to iron feeding.

Table 3 shows the proposed model based on goodness-of-fit indices, which include the chi-square test with an absolute goodness-of-fit index. As the chi-square increases, the fitness of the model decreases. A significant chi-square test reveals a considerable difference between the anticipated and observed covariance. However, due to the inclusion of sample size in the chi-square formula, larger samples often result in inflated values that frequently achieve statistical significance. Consequently, numerous researchers have analyzed the chi-square with its degrees of freedom, referred to as the relative chi-square (CMIN/DF) [33]. In this context, values of 2 or less are typically employed as benchmarks, which are regarded as a standard for assessing model fit. To evaluate the model fit indices, various metrics were utilized, including the parsimonious normed fit index (PNFI), comparative fit index (CFI), parsimonious comparative fit index (PCFI), incremental fit index (IFI), goodness-of-fit index (GFI), and root mean square error of approximation (RMSEA), all of which are detailed in Table 3.

The R2 index shows the amount of explained variance of the endogenous latent variables. The coefficient of determination of the variable of mothers' practices is 0.40, which shows that health literacy and mothers' knowledge can predict 40% of the changes in mothers' practices, which is partial or moderate. Table 4 also shows the standard coefficients of the routes, Fig 1 shows the proposed model of the research, and Fig 2 shows the final model.

**Table 3. The fitness indicators of the proposed model of the current research.**

| Fitness indicators | $x^2$ | df | p value | CMIN/Df | RMSEA($CL_{90\%}$) | PNFI | CFI | PCFI | IFI | GFI | SRMR |
|---|---|---|---|---|---|---|---|---|---|---|---|
| Proposed model | 10.585 | 13 | 0.646 | 0.814 | 0.001(0.000-0.04) | 0.615 | 1 | 0.619 | 1 | 0.993 | 0.018 |

Abbreviations; CMIN/DF: chi-square/degree-of-freedom ratio; RMSEA: root mean square error of approximation; PCFI: parsimonious comparative fit index; GFI: goodness-of-fit index; PNFI: parsimonious normed fit index; IFI: incremental fit index; CFI: comparative fit index.

**Table 4. Standard coefficients of the paths of the final pattern (modified).**

| Paths | β | SE | CR | P value |
|---|---|---|---|---|
| Knowledge → Health literacy | 0.251 | 0.077 | 3.891 | <0.001 |
| Health literacy → Practice | 0.482 | 0.084 | 5.604 | <0.001 |
| Knowledge → Practice | 0.410 | 0.082 | 4.398 | <0.001 |

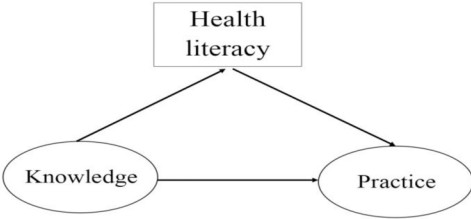

**Fig 1. Hypothesized model in relationships between variables.**

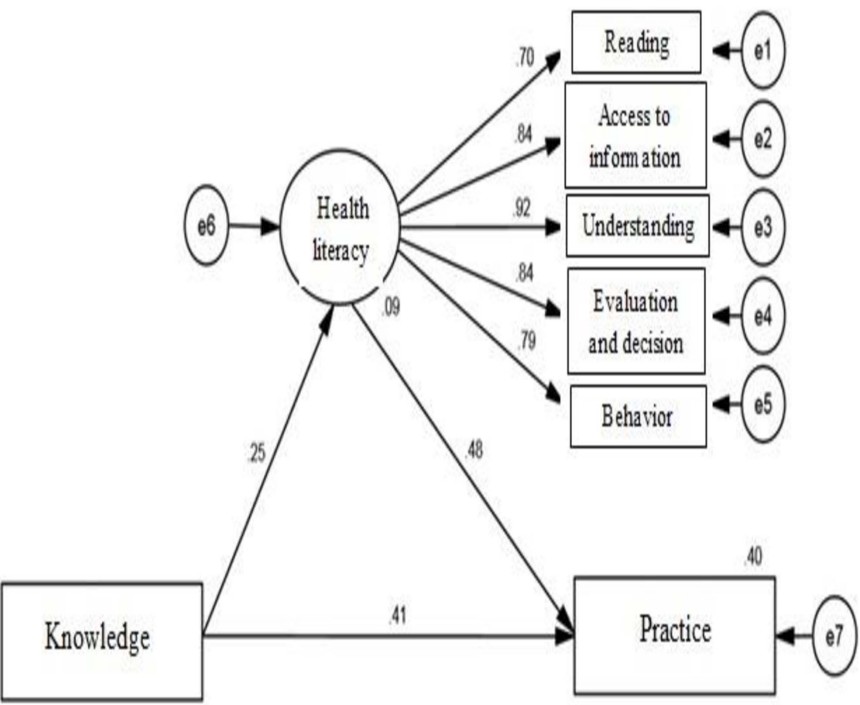

**Fig 2. Standard coefficients of the modified model.**

The outcomes of the mediation analysis conducted through the bootstrap test utilizing AMOS software are presented in Table 5. As shown in the table, the lower limit of the confidence interval for health literacy as a mediating variable between mothers' knowledge and mothers' practices is 0.087, and its upper limit is 0.198. The confidence interval is set at 95%, with 5000 bootstrap resamples conducted. Health literacy serves as a mediating factor in the association between maternal knowledge and their performance

## Discussion

This study sought to explore the connection between maternal knowledge and their practices regarding administering iron drops to children aged 12 to 24 months, while also examining the mediating influence of health literacy. The findings indicated a significant correlation between knowledge, practice, and health literacy, with health literacy as a mediating factor in this relationship. 54.7% of the mothers had a moderate level of iron feeding, and this finding was similar to that of other studies [28,34]. A recent report from Iran mentioned that approximately 63% of mothers had low performance [35].

Another study revealed a relationship between mothers' knowledge and performance in iron-feeding children. Interestingly, a study reported that the prevalence of iron deficiency anemia is high among children aged 6 to 12 months, and one of the reasons is the lack or

**Table 5. Bootstrap results for the indirect path of the revised model.**

| Paths | Indirect effect | Upper limit | Lower limit |
|---|---|---|---|
| Mothers' knowledge to the practice through health literacy | 0.121 | 0.087 | 0.198 |

deficiency of iron supplementation [36]. Therefore, these findings support the theory that the intention to perform a health practice is related to health literacy, and people's awareness is related to their health performance.

Caregivers are essential in the management and treatment of childhood illnesses. Given that most children are unable to care for themselves, the timing of interventions and the quality of care provided are largely influenced by the actions of the caregiver, which ultimately shape the prognosis of the illness [37].

The findings of this study provide valuable insights into the use of iron supplements by mothers for children aged 12 to 24 months and the mediating role of health literacy in this context. The results indicated that a significant proportion of mothers in the study population did not adequately supplement their children with iron; however, there are conflicting results in another study. Hironaka's study [38] revealed that caregivers with limited health literacy had better performance in giving supplements, and these results deserve further discussion [39,40].In the present study, a relationship between demographic characteristics such as parents' occupation and the adequacy of income and performance was observed, which is in line with other studies [35,41–43]; however, another study did not find a significant relationship between mothers' performance and their employment status, but generally, the practices of employed mothers were better than those of housewives [28]. Studies have reported that the level of health literacy is influenced by factors such as occupation, income, and education [44–46]. The findings align with various research efforts, including the work of Berkman et al., which demonstrated that health behavior is shaped by a range of factors mediated by health literacy (HL) [39]. Additionally, Manganello's study indicated that health behavior is affected both directly and indirectly by multiple factors through the lens of HL [40].

In Iran, mothers can access iron supplements free of charge; however, it is possible that additional elements, including cultural influences and the quality of the prescribed iron supplements, play a significant role in shaping mothers' iron supplementation practices.

To the best of our knowledge, this is the first study to demonstrate the significance of pathways of health literacy on the knowledge and practice of iron supplementation among children. Moreover, the mediating role of health literacy emerged as a key factor influencing the use of iron supplements by mothers. Higher levels of health literacy were associated with an increased likelihood of adequate iron supplementation for children. Furthermore, the study identified several related factors that may influence the uptake of iron supplements by mothers for their children.

The implications of these findings are complex and multifaceted. They underscore the importance of targeted educational programs aimed at improving public awareness of dietary supplements, particularly among mothers. Healthcare practitioners play a crucial role in guiding mothers toward informed and safe choices concerning dietary supplements, emphasizing the need for open communication and cooperative decision-making strategies.

## Limitations of the study

Firstly, the cross-sectional nature of the study avoids the assessment of causality relationships among variables. Secondly, recall bias, can lead to either overestimation or underestimation of associations, ultimately affecting the study's validity and generalizability. The influence of modifiers was unable to be controlled. The last one, samples were taken from selected clinics in Tehran, and subjects' characteristics may differ from other districts, hence the results may not be generalizable to the broader population. We suggest future studies with higher sample sizes and by random selection from different social classes of the population.

## Conclusion

A notable correlation was identified between health literacy and both knowledge and practices concerning iron supplementation; moreover, health literacy might mediate mothers' knowledge and practices. Therefore, by promoting health literacy, it is possible to improve the practice of iron feeding by mothers, which is an important factor in controlling IDA. Considering the effect of anemia on children's health, paying attention to mothers' health literacy as an important factor in improving their performance is essential.

### Application of the study

Considering the relationship between knowledge and practice of Iron supplementation and health literacy mediating role showing the need to develop educational programs to increase the knowledge and health literacy to improve Iron supplementation in children less than 2 years and reduce the prevalence of anemia. In addition developing tailored health literacy interventions for underperforming groups.

### Acknowledgments

We acknowledge all the mothers who participated in the study.

### Author contributions

**Conceptualization:** shabnam Omidvar.

**Data curation:** Ghazal Afshari.

**Formal analysis:** Mohammadreza Kordbagheri.

**Investigation:** shabnam Omidvar.

**Methodology:** shabnam Omidvar, Mohammadreza Kordbagheri.

**Software:** Ghazal Afshari.

**Supervision:** shabnam Omidvar.

**Validation:** Mohammadreza Kordbagheri.

**Writing – original draft:** shabnam Omidvar.

**Writing – review & editing:** Ghazal Afshari, shabnam Omidvar, Mohammadreza Kordbagheri.

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
