## [Decision Letter · Decision Letter 0]

19 Dec 2024

Dear Dr. Omidvar,

Thank you for submitting your manuscript to PLOS ONE. After careful consideration, we feel that it has merit but does not fully meet PLOS ONE’s publication criteria as it currently stands. Therefore, we invite you to submit a revised version of the manuscript that addresses the points raised during the review process.

We look forward to receiving your revised manuscript.

Kind regards,

Ahmad Neyazi

Academic Editor

PLOS ONE

Additional Editor Comments (if provided):

Reviewers' comments:

Reviewer's Responses to Questions

**Comments to the Author**

1. Is the manuscript technically sound, and do the data support the conclusions?

Reviewer #1: Yes

Reviewer #2: Yes

2. Has the statistical analysis been performed appropriately and rigorously?

Reviewer #1: Yes

Reviewer #2: Yes

3. Have the authors made all data underlying the findings in their manuscript fully available?

Reviewer #1: Yes

Reviewer #2: Yes

4. Is the manuscript presented in an intelligible fashion and written in standard English?

Reviewer #1: Yes

Reviewer #2: Yes

Reviewer #1: The manuscript is well-written and presents important insights. To further enhance its quality and impact, the following suggestions are offered:

1. Consider incorporating information on health literacy, such as mental health literacy (e.g., DOI: 10.56101/rimj.v3i1.91) and literacy related to specific diseases (e.g., DOI: 10.56101/rimj.v3i1.75), from other countries. This addition would make the literature review more comprehensive.

2. Clearly outline how this study builds on existing research and addresses knowledge gaps, particularly regarding the mediating role of health literacy in mothers' knowledge and practices of iron supplementation.

3. Broaden the discussion of findings in comparison to previous studies. Highlight any unique insights or significant discrepancies to provide a richer context.

4. Expand on the implications of health literacy for broader public health strategies, especially in the context of other nutritional interventions.

5. Include a more detailed explanation of the study’s limitations, specifically addressing how recall bias and selection bias (resulting from the convenience sampling method) may have influenced the findings.

6. Offer specific suggestions for future research that could address these limitations and advance the field.

7. Provide more actionable recommendations based on the findings, such as developing tailored health literacy interventions for underperforming groups.

8. Enhance the clarity of tables and figures by including more descriptive captions and offering detailed interpretations in the text.

9. Elaborate on the sample size calculation and explain how it ensures sufficient statistical power for all analyses performed.

10. Expand the discussion of questionnaire validation processes beyond Cronbach’s alpha, describing any pilot testing or iterative refinements.

11. Offer a more detailed explanation of the statistical methods used for mediation analysis, emphasizing their rigor and the robustness of the findings.

12. Discuss in greater depth how the sample reflects the broader population and what this means for the generalizability of the study’s conclusions.

13. Ensure consistent use of technical terms and minimize redundancy, particularly in the Introduction and Results sections, to improve readability and focus.

Reviewer #2: This manuscript is well-written and demonstrates a high standard of scholarly work. It meets the criteria for publication and is suitable for acceptance in its current form without the need for further revisions.

**Do you want your identity to be public for this peer review?** For information about this choice, including consent withdrawal, please see our Privacy Policy

Reviewer #1: No

Reviewer #2: No

---

## [Author Response · Author response to Decision Letter 1]

31 Dec 2024

All the editor and reviewers' comments have been addressed properly. I prepared a file namely point by point response and attached as a file.

---

## [Decision Letter · Decision Letter 1]

10 Feb 2025

The role of health literacy in the relationship between mothers’ knowledge and practices of iron supplementation in children (aged 12 to 24 months): A structural equation model

PONE-D-24-36544R1

Dear Dr. Omidvar,

We’re pleased to inform you that your manuscript has been judged scientifically suitable for publication and will be formally accepted for publication once it meets all outstanding technical requirements.

Kind regards,

Ahmad Neyazi

Academic Editor

PLOS ONE

Additional Editor Comments (optional):

Thank you for revising the manuscript. After careful consideration of the reviewers' comments, I am pleased to inform you that your manuscript has been accepted for publication.

Reviewers' comments:

Reviewer's Responses to Questions

**Comments to the Author**

Reviewer #1: All comments have been addressed

Reviewer #2: (No Response)

2. Is the manuscript technically sound, and do the data support the conclusions?

Reviewer #1: Yes

Reviewer #2: (No Response)

3. Has the statistical analysis been performed appropriately and rigorously?

Reviewer #1: Yes

Reviewer #2: (No Response)

4. Have the authors made all data underlying the findings in their manuscript fully available?

Reviewer #1: Yes

Reviewer #2: (No Response)

5. Is the manuscript presented in an intelligible fashion and written in standard English?

Reviewer #1: (No Response)

Reviewer #2: (No Response)

Reviewer #1: Thank you for this outstanding work. I believe it is now ready for publication in th PLOS One journal.

Reviewer #2: Thank you for sending the manuscript again for review. The current revision is much better and acceptable for being published.

**Do you want your identity to be public for this peer review?** For information about this choice, including consent withdrawal, please see our Privacy Policy

Reviewer #1: **Yes: ** Abdul Qadim Mohammadi

Reviewer #2: **Yes: ** Nosaibah Razaqi

---

## [Editor Report · Acceptance letter]

PONE-D-24-36544R1

PLOS ONE

Dear Dr. Omidvar,

I'm pleased to inform you that your manuscript has been deemed suitable for publication in PLOS ONE. Congratulations! Your manuscript is now being handed over to our production team.

Kind regards,

on behalf of

Dr. Ahmad Neyazi

Academic Editor

PLOS ONE